# Economic convergence in a globalized world: The role of business cycle synchronization

**Andres Lopez** [1]⊕*, **Sonia De Lucas** [2]⊕, **Maria Jesus Delgado** [1]⊕

**1** Department of Applied Economics, Universidad Rey Juan Carlos, Madrid, Spain, **2** Department of Applied Economics, Universidad Autonoma, Madrid, Spain

⊕ These authors contributed equally to this work.
* andres.lopez@urjc.es

**Data Availability Statement:** GDP data and globalization data can be found in the following URLs: https://www.conference-board.org/data/economydatabase/ https://kof.ethz.ch/en/forecasts-

## Abstract

Increasing economic integration and global synchronization can be key for countries aiming to catch up in GDP per capita terms. Little attention has hitherto been placed in synchronization as determinant of convergence. In this paper we estimate the effect of economic globalization and synchronization on income convergence for a sample of 89 developed and developing countries in the period 1970–2015. We use a dynamic factor model and panel data techniques to undertake the objectives of the paper. We show that synchronized countries (those correlated with the factor) exhibit a higher response on GDP per capita growth with variations on the global business cycle. This implies that synchronization improves growth for that group in global expansionary phases, but also implies risks during global recessions. On the contrary, the effect on growth of an economic globalization index is less relevant for synchronized countries than for asynchronized countries. The latter result implies that asynchronized countries can benefit more increasing their levels of economic globalization.

## 1. Introduction

In the last decade the economic convergence literature has deepened on the debate of why some countries grow faster than others. During this time, the causes of the differences in economic development across countries have been quite diverse, ranging from the commonly accepted view of the institutional factors explained in [1] and [2] to others such as differences in investment levels in human capital as in [3], the rate of innovation or the technological potential. Additionally, the *catch-up effect* from the seminal paper of [4] is hitherto widely accepted: countries departing from relatively low levels of GDP per capita tend to grow faster than richer countries, which will eventually make the former converge in economic terms. However, papers departing from this argument have not always included causal relations determining the relationship of GDP levels and growth. Conditional convergence literature has expanded the search of determinants of convergence, identifying potential variables influencing the catching-up process.

Therefore, focusing on the economic convergence literature, the hypothesis of beta convergence has been refined from the seminal papers which departed from the idea of closing gaps

and-indicators/indicators/kof-globalisation-index.html.

**Funding:** The author(s) received no specific funding for this work.

**Competing interests:** The authors have declared that no competing interests exist.

in GDP as a *natural* mechanism, but did not always dig into the causes of why this process took place. The concept of beta convergence has evolved to conditional convergence, which establishes that countries converge in economic terms *only if* they fulfill certain conditions, such as the previously mentioned: institutional quality, technological change—[5]- or adoption, a given rate of innovation or the scope of human capital. Our paper is embedded in this field of research, as we consider that convergence is conditioned by country levels of economic globalization (economic integration) and the degree of business cycle synchronization, expressed as a correlation with a global business cycle. One novel aspect of this approach is the dynamic implications that these two variables display: for a given country, the economic development process is usually associated with increasing levels of globalization, and as we try to show, also with a higher synchronization with the global business cycle.

The study of convergence across countries is a recurrent topic in economics, but it has reemerged with strength on the political debate from the 2008 crisis, especially inside specific geographic areas or convergence clubs such as the European Union [6, 7] where convergence may have suffered a sudden stop. The causes and consequences of recent episodes of divergence in GDP per capita levels globally and within economic blocs are again at the center of the research and policy discussions, as well as the pros and cons of economic integration, with literature estimating the economic effects for winners and losers from new trade agreements, custom unions or other types of integration schemes already in place. This debate may become increasingly important once the COVID-19 recession (the Great Lockdown) is overcome, since the very internationally integrated global supply chains have been at the center of the supply constraints experienced in medical devices and sanitary products.

At the same time, synchronization of business cycles across countries seems essential for the correct development of a process of economic integration such as the European Union. Economic policy coordination across countries (European Semester) should help in the task of synchronization and also the specific tools developed to minimize or prevent the adverse effects derived from asymmetric shocks (for example, the European Stability Mechanism and the Banking Union). Analogously and more broadly, if we understand globalization as a global economic integration process, synchronization of business cycles across countries should be critical for the successful implementation of policies aimed to increase globalization, and both globalization and synchronization should play a role in economic convergence.

It has been argued that countries take policy decisions that imply varying degrees of integration with the rest of the world for every component of economic globalization: trade, finance and migration. From a historical economic perspective, and more pronouncedly since the beginning of the 1990s, developing countries are increasing globalization levels at a faster pace than developed ones, both financially and in terms of trade (see [8]). At the same time, for a given country, higher levels of globalization tend to imply a higher degree of synchronization with the fluctuations of the World economy, which theoretically should refine the formation of a global business cycle. But these two can be interpreted as strong assumptions. From the trade channel, the increasing degree of specialization linked to economic development could result in asymmetry of the shocks across countries in spite of trade integration. From the financial channel, countries may become more vulnerable to negative external shocks in the process of capital account liberalization [9]. Therefore, synchronization and the formation of a global business cycle could be hindered from both channels. Our paper also aims to shed some light on this open issue: to strengthen the previous assumptions, in the data section we will analyze the relation between globalization, GDP per capita growth and synchronization, and will discuss the stylized facts that the data present.

In our view, with this paper we are filling a gap in the literature, since the convergence issue has not yet been approached from the synchronization perspective. In this respect, the main

novelty of the paper emanates from the country grouping technique: the dynamic factor used to select synchronized countries. This idea contributes to the existing literature by adding a condition for income convergence. As we will point out later, we observe that asynchronized countries rarely rank among the highly developed, and the group of synchronized is composed mainly only by middle income and upper income countries.

Focusing on our empirical specification, we argue that economic globalization levels (globalization, for simplicity) and the synchronization of country specific cycles with the global business cycle (synchronization, for simplicity) are among the main determinants of convergence. Specifically, we aim to prove that *synchronization improves convergence in global expansionary phases but implies risks during global recessions*. The second goal of the paper is to determine the role that globalization plays in the convergence process: we demonstrate that *asynchronized countries show a stronger effect on growth with own increases of globalization levels*. The latter hypothesis brings in mind the absolute beta convergence argument, since asynchronized (developing) countries exhibit a higher response to growth than the sample comprising only synchronized (developed) countries, but we add the conditional distinction arising with differences in synchronization.

The main purpose of this paper is therefore to empirically estimate the effects of globalization on economic convergence, and how these effects vary with synchronization. Another purpose, as we have previously discussed, is to establish a methodology to select the subgroup of synchronized countries, and compare their behavior in convergence terms with our full sample comprising economies of different levels of development. For that we build a dynamic panel data model, using as explanatory variables an economic globalization index (the economic component of the *KOF index*) representing globalization levels for each country and a dynamic factor capturing common fluctuations of GDP series. We use this dynamic factor to replicate the behavior of a global business cycle, since we consider it the best expression of the short-term development of globalization.

Our empirical specification quantifies both effects on global convergence for a sample of 89 developing and developed economies. We compare the results with a second set of coefficients for a subsample of 49 synchronized countries. Synchronization is defined as a high enough correlation with the factor (i.e., with the global business cycle). Therefore, the factor is not only an explanatory variable in the convergence model: it is also used as a country grouping technique.

The rest of the paper is structured as follows. In section 2 we review the literature and describe the theoretical relationship implied in our convergence model. In section 3 we describe the data and establish the econometric methodology. In section 4 we show the results for the full sample and the subsample of countries and discuss the estimations of both groups. Finally, in section 5 we present our conclusions and point to possible extensions of our research.

## 2. Literature review and theoretical framework

As [10] put it, the essence of globalization is the participation of many individual countries in a world business cycle. Also, and as has been previously exposed, a popular view is that increasing globalization implies more synchronization of business cycles across countries, and should contribute to the formation of a global business cycle.

Our aim is to use from the economic convergence point of view the literature relating globalization and growth and another strand of literature relating business cycles and growth, approaching the latter from a global perspective and putting special attention to the role that synchronization plays. To our knowledge there is no paper estimating both effects on

convergence. In this section we review the literature on both fields of research, trying to find common ground for the development of the theoretical relation implied in our model.

## 2.1. Effect of globalization on growth

Our paper is mainly related to the strand of literature relating the effects of globalization on growth, for which we depart from the paper of [11] where the globalization index we use in this paper is developed and its effect on growth is calculated. Previous literature extracts conclusions depending on which of the classical three economic dimensions of economic globalization: trade [12, 13], labor markets and financial markets [12, 14] or other indicators of globalization (ideas, political and social globalization as in [2]). In this respect, some components, mainly the trade and financial ones, show different outcomes on growth depending on the period or country groups upon study. Although the financial channel plays a fundamental role in determining how shocks propagate across countries, it does not fully explain the impact of globalization on the economy. Other observable components of economic globalization, such as the trade channel, may show effects of the same or different sign. With the index used in this paper we isolate the effect of globalization in a single variable, trying to capture its whole level's effect on growth.

More recently, additional literature has explored the relationship between globalization levels and income convergence using as an explanatory variable the same globalization index that we use in this paper. [2] estimate the effect of its variation on the rate at which a country closes the GDP per capita gap with the richest country (what in the literature is called the *horizontal convergence rate*), controlling for political and institutional quality. They conclude that increasing globalization promotes income convergence, although they use as explanatory variables not just the economic component of the globalization index, but also the social and the political ones. [15] investigate the effect of globalization on income convergence in a very similar model to the one presented in this paper. They decompose the effect on growth for every component of the index. They conclude that, generally, globalization has promoted growth. The paper also studies sigma and beta convergence for globalization levels instead of income, and concludes that globalization convergence has occurred across countries for every component of the index in the period studied.

Following the above, the division of the impact of globalization in more than one explanatory variable may be useful in determining the channels through which economic shocks propagate internationally. However, its overall effect on growth, and therefore on income convergence, is better captured using a weighted index of globalization. Additionally, we avoid the collinearity problems that may arise using the sub-dimensions of globalization as different explanatory variables.

## 2.2. Effect of global fluctuations on growth

As has been previously discussed, countries aiming to increase economic integration with the rest of the world are expected to increase synchronization with the global economy. In this subsection we review literature describing how global fluctuations affect country specific fluctuations, and the approaches developed to capture global fluctuations that are in line with the one used in this paper.

The literature relating business cycles and their effect on growth has generally been built on the assumption that shocks domestically generated are the ones affecting growth, as suggested in [16]. In this paper we assume that the main source of country specific economic fluctuations is the global business cycle, especially, as it seems obvious, for countries more synchronized with this cycle. [17] conclude that a significant portion of G-7 output fluctuations are driven

by a common world indicator, especially during contractions. [18] also estimate the effects of global factor shocks in the economic activity of 36 small open economies, and find a negative and persistent effect. The model of [19] assesses the global, regional and country-specific factors explaining business cycles, and finds that regional factors have increased and the global factor has decreased in importance since the mid-1980s. Some other authors have questioned the existence of common economic factors causing global fluctuations. For example, in [20] the authors claim that papers referring to both world and European business cycles are previously assuming the existence of a common driving factor, and conclude that it cannot be taken for granted for European economies. [21] results suggest the non-existence of a global business cycle, but an increase in common regional fluctuations in the period studied.

One of the main problems when estimating global effects on countries GDP growth is that global cycles are difficult to define, as GDP data is not accurate enough for some countries. The definition of global recessions is also not clear, both theoretically and empirically. Negative growth rates in world GDP are difficult to observe (at least until COVID-19 recession) since national GDP series across countries are not enough synchronized, and negative growth rates in some countries compensate with positive growth rates in others. A solution for this limitation is to find a significant set of countries to extract their common GDP behavior in the short run and use this common information to simulate global fluctuations. In the spirit of [19] we propose a dynamic factor model to use as a proxy of this global cycle.

As has been argued by [22] global factors containing common behavior of country economic cycles tend to explain a higher share of developed economies cycles. Analogously in our paper, once estimated the global business cycle we estimate its effect on growth and determine that it is higher for a subsample of synchronized countries.

From this section we can argue that our two main explanatory variables are closely related both theoretically and empirically. As we will discuss further in the next section, more economically integrated (globalized) countries present more business cycle synchronization. But rather than measuring the effect of globalization on the synchronization of business cycles, our goal is to estimate the effects of globalization levels and global economic fluctuations on economic convergence, and how these effects vary with synchronization.

## 2.3. Beta-convergence and other convergence approaches

Our research is not including methodological novelties in the convergence literature, only the above mentioned new explanatory variables included in the model. However, we consider appropriate to review the main papers covered to develop our convergence model, and to briefly mention other methodologies used to approach income convergence across countries or regions.

To develop our convergence model we depart from the widely known methodology of [4], using an updated version of the model as presented in [23]. We add to our dynamic panel data model two additional explanatory variables (the global business cycle that we estimate and an economic integration indicator). Income convergence is measured with own calculation of speed of convergence, and by interpreting the coefficients of our explanatory variables, that have an effect on economic growth.

Other methodologies could have been used to measure income convergence. For instance, [2] used the horizontal convergence rate, to measure the rate at which a country closes the GDP per capita gap with the richest country. This methodology is very useful to compare income convergence across countries (or in a pairwise procedure), since it uses the same reference (the richest country of the group) to calculate the rate of convergence. We consider our income convergence specification to be more useful to compare convergence between groups

**Table 1. Overview of reviewed sources.**

| Authors | Year | Summary points |
|---|---|---|
| Nell, K.S. | 2020 | • Absence of conditional convergence in the post-1989 globalization period |
| Gygli, S., Haelg, F., Potrafke, N. et al. | 2019 | • Updated version of the KOF index |
| Harger, K., Young, A. T. and Hall, J. | 2017 | • Effect of globalization and institutional quality on income convergence. |
| Villaverde, J. and Maza, A. | 2011 | • Effect of globalization on growth and convergence |
| De Lucas, S., Delgado, M. J. and Alvarez, I. | 2011 | • Factor models for identification of countries sharing international cycles |
| Artis, M. and Okubo, T. | 2008 | • Globalization and business cycle transmission |
| Dreher, A. | 2006 | • Development of the KOF index<br>• Effect of globalization on growth |
| Fatás, A. | 2002 | • Effect of business cycles on growth |

of countries, as it is our case, since we are measuring average effects on economic growth in the dynamic panel, and the inclusion of a high-income country in a given group should not influence the results as much as it would if we used the horizontal convergence methodology. Other possibly useful methodological approach is a probabilistic definition of convergence, that [24] uses to investigate long-run income convergence in the U.S. Properties of all regional income series pairs are studied with unit root and co-integration tests, and they difference between strong and weak convergence. In Table 1 we briefly summarize some papers that have inspired our research.

## 3. Data and econometric methodology

In this section we describe the data used in this paper and present the econometric methodology. We also make a descriptive analysis extracting some stylized facts of the data, which are useful to understand the econometric model and the theoretical relationship between the variables.

Our methodological approach aims to study convergence on a global scale, using a traditional economic convergence model. We highlight two main novel aspects. First, we use synchronization as a means to group countries with a common cyclical behavior to segment our sample, and compare the convergence results with this subgroup and all the countries upon study. Second, the two explanatory variables explaining the globalization effect on growth: the common global fluctuation effect is captured with a dynamic factor and the globalization levels effect is captured with an index.

### 3.1. Data

To estimate our model we depart from the following two databases from which we obtain country GDP data and an index of economic globalization for every country.

The GDP data is extracted from The Conference Board Total Economy Database™, April 2019. We use a balanced panel containing yearly data from 1970 to 2015 of a total of 89 developing and developed countries. In order to obtain a balanced panel, we use data for the 89 countries for which the globalization index data was available in 2016 for the full data range in the regions studied: North America, Latin America, Western Europe, Eastern and Central Europe, Pacific and Africa and the Middle East. We proceed in two steps: First, we transform the data in natural logarithms and take the first differences to estimate the dynamic factor. We estimate the dynamic factor using Grocer Econometric Toolbox of [25] from *Scilab*. With the

factor we extract the common fluctuation of GDP across countries in the sample. Second, we compute the convergence coefficients in a panel regression using the common global fluctuation, the globalization index and country GDP levels as explanatory variables.

We get economic globalization levels yearly data from *The KOF Globalization Index*. This indicator is a useful measure of the overall degree of economic globalization of a country since we avoid the multicollinearity problems that may arise when including more than one sub-index of globalization as explanatory.

The version of the index we use is composed of a set of 23 variables. The version of the index used in this paper is composed of 23 variables measuring the different components of globalization, see [11]. An updated index composed of 43 variables can be used, see [26]. We use for our estimations only the economic globalization component, which is measured with two indexes: the first includes actual flows (trade, FDI, portfolio investment and income payments to foreign nationals) and the second measures trade and capital flow restrictions. These two are aggregated in an overall economic globalization sub-index.

Before presenting the model, we consider it important to highlight two stylized facts of the data we work with, which have implications for our methodology and are useful to put into context the results. The first stylized fact concerns the relationship between globalization and synchronization. As we will see, synchronized countries tend to rank among the highly globalized (Table 2A). The relation seems stronger for asynchronized countries, which are rarely ranked among the highly globalized. With the distinction of asynchronized and synchronized countries we aim to differentiate the effects of globalization and synchronization on economic convergence for developing and developed countries, respectively, and the economic implications of these different effects. We additionally show, using a scatter plot, the positive relation existing between synchronization and economic globalization (S3 Fig). The second stylized fact aims to capture the static and dynamic relations between GDP levels and synchronization (S1 Fig and Table 2B). Synchronized countries seem to have performed relatively better in terms of growth, but this effect has vanished from the financial crisis of 2008, which supports our argument that synchronized countries face higher risks from contagion from global crises.

**Table 2. A. Economic Globalization levels and synchronization.** B. GDP per capita levels and synchronization.

A.

|  | Synchronized countries (49) | asynchronized countries* (40) |
|---|---|---|
| **Among the 40 more globalized in the world** | 21 | 0 |
| **Among the 50 more globalized in the world** | 26 | 1 |
| **Among the 80 more globalized in the world** | 33 | 6 |
| **Among the 100 more globalized in the world** | 37 | 8 |

B.

|  | Synchronized countries (49) | asynchronized countries* (40) |
|---|---|---|
| **Among the 40 with higher GDPpc in the world** | 27 | 1 |
| **Among the 50 with higher GDPpc in the world** | 31 | 3 |
| **Among the 80 with higher GDPpc in the world** | 43 | 12 |
| **Among the 100 with higher GDPpc in the world** | 47 | 19 |

*Note*: KOF Economic globalization levels for full (World) KOF database in 2015.

GDPpc levels in 2015, data from the Conference Board Database.

*asynchronized countries: Panel A countries (89)–Panel B countries (49). See list of synchronized and asynchronized countries in Appendix 1.

As has been previously discussed, a novel aspect of this paper is how the dynamic relation of the variables upon study affects our results. To some extent, the relation of the table above points to the idea that over time, for a given country, increasing globalization levels would increase its synchronization. As depicted in Table 2A, 37 of the 49 synchronized countries from our study are among the 100 more globalized countries in the World in 2015 following data from the KOF index, 33 are among the 80 more globalized and 26 of them are among the 50 more globalized. On the contrary, none of the 40 asynchronized countries are among the 40 more globalized, only 1 is among the 50 more globalized and 6 of them are among the 80 more globalized. It is particularly interesting to point that the relation between synchronization and high levels of globalization seems weaker than the relation between asynchronization and low levels of globalization: it is extremely odd to find highly globalized countries among the asynchronized. Therefore, *the globalization level works as a necessary condition for convergence*: if a country has not reached a minimum globalization threshold, then it is not synchronized (developed). Although this does not imply any causal relation, from the previous distribution of countries we can figure out that our sample of synchronized countries is comprised of mostly developed countries, and the 40 asynchronized from the full sample (89) are mostly developing.

To support the previous argument and to give strength to our implicit assumption that asynchronized countries tend strongly to be ranked among the developing ones, Table 2B depicts the distribution of synchronized and asynchronized countries depending on their level of development, proxied by GDP per capita in the year 2015. As can be seen, a clear relation exists since only 1 of the asynchronized countries ranks among the 40 more developed. Analogously, 47 of the 49 synchronized countries are ranked among the 100 more developed.

Among the previous studies relating globalization and business cycle synchronization, there is a strand of literature widely accepting that for a given country a higher level of globalization implies more synchronization. Some of these find openness in trade and finance as indicators of the transmission of business cycles and their synchronization across countries (for example, [10]). Departing from this argument, more economic integration (more globalization) goes hand in hand with more common fluctuations. Therefore, synchronized countries are expected to be more globalized, as can be extracted by comparing the synchronization and globalization levels across countries in the previous table. Contrary to this, [27] find evidence of financial integration de-synchronizing national outputs from the global business cycle. [28] estimates the combined effect of trade and a varying degree of financial integration in a multi-sector setting, concluding that trade integration enhances synchronization and financial integration reduces it. These papers although presenting different results make clear the close relationship between our two main explanatory variables, as financial and trade integration (i.e., economic globalization) have an effect on the degree of synchronization of country specific cycles with the global business cycle. However, to our knowledge, this discussion has not been used to explore different patterns of convergence in an empirical model and we aim to fill this gap.

An additional fact that we aim to discuss is the dynamic relation between synchronization and GDP growth. As shown in S1 Fig, synchronized countries tend to grow more on average until the global financial crisis of 2008, when the trend reverses. From that moment on, asynchronized countries converge on average GDP index in per capita terms. This would support our hypothesis that synchronized countries face more risk from contagion in global recessions.

S2 Fig is illustrative of the globalization process in economic terms. Levels are shown for high, middle (upper and lower) and low-income countries from 1970 to 2015. Some countries, mainly in Asia, have converged in terms of globalization with the Western countries. We aim

to demonstrate that this convergence in globalization (in economic globalization, to be precise) has translated into GDP convergence.

As for the close relationship between economic integration and business cycle synchronization with the global business cycle, the following scatter (S3 Fig) shows it graphically. We plot the 49 synchronized countries from our estimates of the factor loadings representing synchronization, and the economic component of the globalization index (value of 2015) as a proxy for economic integration.

## 3.2. The dynamic factor

Our methodological approach starts with the use of a dynamic factor to estimate the common fluctuation pattern in the set of country GDP series $X_t = \Delta \ln GDP_{i,t}$, with a double objective:

1. The selection of countries sharing the behavior of the global business cycle extracted with the factor. In this respect, a novel aspect of the paper is that we use the factor as a country grouping technique.

2. The use of the global business cycle as an explanatory variable in the convergence model to evaluate its effect on convergence across countries. The factor series is included in the panel $i$ times to simulate the effect of the global business cycle across countries.

Following the methodology proposed by [29], where the latent factors follow a time series process commonly taken as a vector autoregression (VAR), the formal dynamic factor model is:

$$X_T = \Lambda_T F_T + e_T \tag{1}$$

$$f_T = \psi(L)f_{t-1} + \eta_t \tag{2}$$

where there are N countries, so $X_T$ and $e_T$ are N×1, there are m dynamic factors so $f_T$ and $\eta_T$ are m×1, $\Lambda = (\lambda_1,\lambda_2,\ldots,\lambda_m)$ is N×m, $L$ is the lag operator, and the lag polynomial matrix $\psi(L)$ is m×m. The i-th $\lambda_i$ are called factor loadings for the i-th countries, $X_{i,t}$.

The idiosyncratic disturbances, $e_T = (e_{1,T}, e_{2,T},\ldots,e_{N,T})'$ are the specific elements of each series contained in a vector; they are serially correlated and slightly cross-sectionally correlated with other variables in the model and are mutually uncorrelated at all leads and lags, that is, $Ee_{i,t}e_{j,s} = 0$ for all $s$ if $t \neq s$. They are assumed to be uncorrelated with the factor innovations at all leads and lags, that is, $Ee_t \eta'_{i-k} = 0$ for all $k$. The $pth$ order autoregressive polynomial, $\psi_i(L)$, is assumed to have stationary roots.

The standard estimation method is provided by maximizing the likelihood of the corresponding model and estimation accuracy via the Kalman filter, after a suitable reparameterization of the model in state-space form, assuming that all the processes in (1)-(2) are stationary and not cointegrated (see [30] for details). We use the GROCER's Econometric Toolbox [31].

The existence of only one common factor $f_{1,t}$ is confirmed employing the statistical criterion proposed by [32].

## 3.3. The convergence model

Our model departs from the theoretical representation of the beta convergence process presented in [23]. From this equation natural log of GDP per capita of a region $i$ depends on the following relation:

$$\ln(y_{i,t}) = \beta_0 + (1 - \beta_1)\ln(y_{i,t-1}) + u_{i,t} \tag{3}$$

It is straightforward to transform (3) into the beta convergence equation:

$$\ln\left(\frac{y_{i,t}}{y_{i,t-1}}\right) = \beta_0 - \beta_1\ln(y_{i,t-1}) + u_{i,t} \tag{4}$$

where $0 < \beta_1 < 1$ implies a negative relationship between GDP per capita growth and initial log GDP per capita, and $u_{i,t}$ has mean zero and a finite variance and is independent over $t$ and $i$ as shown in [23].

The panel data convergence model is presented in a growth accounting framework as established in [33]. We extend (4) with the globalization variables, plugging in the previous equation the global business cycle (factor) and also the economic globalization index (KOF):

$$\ln\left(\frac{y_{i,t}}{y_{i,t-1}}\right) = \beta_0 - \beta_1\ln(y_{i,t-1}) + \beta_2(Factor_{i,t}) + \beta_3(KOF_{i,t}) + \alpha_i + u_{i,t} \tag{5}$$

where $\beta_1$ is the neoclassical (beta) convergence parameter, $\alpha_i$ is the fixed effects coefficient and $\beta_2$ and $\beta_3$ represent respectively the effects of the global business cycle and the globalization index.

Due to the potential reverse causality we run fixed effects two-stage least squares for our balanced panel data (see [34] for details of the two-stage estimation, using the ivpanel function from the Panel Data Toolbox in *Matlab*). With the inclusion of an instrument in the regression we account for potential endogeneity of the explanatory variables when estimating the model.

With the fixed effects coefficients we estimate the effect of the explanatory variables within each country over time, controlling for unobserved heterogeneity [35]. Individual effects are correlated with each explanatory variable, so in the fixed effects estimation explained and explanatory variables are demeaned to compute the within estimators. These estimators are unbiased and consistent for $n \rightarrow \infty$ (see [34] for details). All time invariant factors are eliminated with this transformation, including the non-observed fixed effects. Intuitively, the estimated coefficients will tell us the effect of variations around the mean of the explanatory variables on variations around the mean of the explained one. We estimate the parameters of the panel data model with our full sample of 89 countries and with a sub-sample of synchronized countries. Our first hypothesis is that synchronization improves convergence in global expansions but implies risks from contagion in global recessions. The second hypothesis is that asynchronized countries can benefit relatively more from increasing levels of globalization.

## 4. Results and discussion

In this section we first estimate the global business cycle and identify synchronized countries sharing its behaviour, following [36] methodology. Factor loadings in Table 3 can be interpreted as correlations between the country GDP cycle and the global business cycle that we estimate. Results for the convergence model estimations can be found in columns IV and VIII from Table 4, showing full sample (panel A) and subsample (panel B) results, respectively. From our empirical results we draw that differences in synchronization and globalization levels are the main drivers of economic convergence.

### 4.1. Factor correlations and synchronization

Table 3 shows factor loadings for the subgroup of synchronized countries. To select the synchronized countries, we run a static factor to find common behaviour from the GDP growth data of the full sample of 89 countries. [32] criterion is used to confirm the existence of only

**Table 3. Dynamic international common factor (49 synchronized countries).**

| Countries | Factor loadings | AR parameters | Residual variance |
|---|---|---|---|
| *North America* | | | |
| **Canada** | 0.61 (4.94) | 0.08 (0.68) | 0.65 (5.74)*** |
| **United States** | 0.54 (4.13) | 0.12 (0.99) | 0.75 (5.77)*** |
| *Western Europe* | | | |
| **Austria** | 0.8 (6.97) | 0.3 (2.49)*** | 0.37 (5.55)*** |
| **Belgium** | 0.91 (9.2) | -0.2 (-1.46) | 0.22 (5.06)*** |
| **Cyprus** | 0.32 (2.43) | 0.08 (0.66) | 0.89 (5.81)*** |
| **Denmark** | 0.62 (5.02) | 0.08 (0.64) | 0.66 (5.73)*** |
| **Finland** | 0.84 (7.19) | 0.3 (2.45)*** | 0.36 (5.52)*** |
| **France** | 0.94 (9.36) | 0.25 (1.79)* | 0.16 (4.92)*** |
| **Germany** | 0.76 (6.39) | 0.52 (4.82)*** | 0.4 (5.58)*** |
| **Greece** | 0.55 (4.55) | 0.24 (2.01)*** | 0.57 (5.75)*** |
| **Iceland** | 0.3 (2.19) | 0.2 (1.64) | 0.85 (5.81)*** |
| **Ireland** | 0.37 (2.82) | 0.48 (4.52)*** | 0.65 (5.8)*** |
| **Italy** | 0.71 (6.78) | 0.57 (5.47)*** | 0.29 (5.52)*** |
| **Luxembourg** | 0.48 (3.84) | 0.02 (0.2) | 0.79 (5.78)*** |
| **Netherlands** | 0.78 (6.69) | 0.11 (0.91) | 0.46 (5.61)*** |
| **Norway** | 0.44 (3.52) | 0.46 (4.18)*** | 0.59 (5.77)*** |
| **Portugal** | 0.75 (6.44) | 0.26 (2.16)*** | 0.42 (5.62)*** |
| **Spain** | 0.52 (4.36) | 0.35 (3.00)*** | 0.52 (5.75)*** |
| **Sweden** | 0.92 (7.67) | 0.49 (4.39)*** | 0.33 (5.39)*** |
| **Switzerland** | 0.84 (7.06) | 0.23 (1.89)* | 0.41 (5.55)*** |
| **United Kingdom** | 0.7 (5.47) | 0.32 (2.73)*** | 0.56 (5.69)*** |
| *Central and Eastern Europe* | | | |
| **Bulgaria** | 0.35 (2.5) | 0.22 (1.8)* | 0.86 (5.81)*** |
| **Hungary** | 0.43 (3.12) | 0.3 (2.52)*** | 0.76 (5.79)*** |
| **Romania** | 0.22 (1.9) | 0.61 (6.4)*** | 0.55 (5.81)*** |
| *Oceania and Asia* | | | |
| **Australia** | 0.48 (3.78) | 0.06 (0.49) | 0.78 (5.78)*** |
| **New Zealand** | 0.36 (2.54) | 0.36 (3.13)*** | 0.8 (5.81)*** |
| **Hong Kong** | 0.37 (2.71) | 0.19 (1.56) | 0.85 (5.8)*** |
| **Japan** | 0.56 (5.41) | 0.6 (6.03)*** | 0.34 (5.66)*** |
| **Singapore** | 0.44 (3.26) | 0.42 (3.79)*** | 0.71 (5.78)*** |
| **Indonesia** | 0.3 (2.06) | 0.39 (3.45)*** | 0.86 (5.81)*** |
| **Malaysia** | 0.28 (1.9) | 0.29 (2.47)*** | 0.91 (5.82)*** |
| **Philippines** | 0.33 (2.5) | 0.57 (5.65)*** | 0.67 (5.8)*** |
| **Thailand** | 0.23 (1.65) | 0.39 (3.42)*** | 0.83 (5.82)*** |
| *Latin America* | | | |
| **Argentina** | 0.34 (2.47) | 0.15 (1.24) | 0.89 (5.81)*** |
| **Barbados** | 0.55 (4.95) | 0.51 (4.75)*** | 0.41 (5.7)*** |
| **Brazil** | 0.49 (3.72) | 0.42 (3.82)*** | 0.66 (5.77)*** |
| **Colombia** | 0.35 (2.51) | 0.46 (4.23)*** | 0.76 (5.8)*** |
| **Costa Rica** | 0.27 (2.04) | 0.02 (0.2) | 0.92 (5.82)*** |
| **Guatemala** | 0.31 (2.5) | 0.56 (5.56)*** | 0.6 (5.8)*** |
| **Jamaica** | 0.32 (2.49) | 0.47 (4.35)*** | 0.64 (5.8)*** |
| **Mexico** | 0.51 (3.9) | 0.14 (1.16) | 0.76 (5.78)*** |
| **Trinidad & Tobago** | 0.33 (2.53) | 0.52 (5.01)*** | 0.65 (5.8)*** |

*(Continued)*

**Table 3.** (Continued)

| Countries | Factor loadings | AR parameters | Residual variance |
|---|---|---|---|
| Venezuela | 0.29 (2.12) | 0.4 (3.63)*** | 0.77 (5.81)*** |
| *Middle East and Africa* | | | |
| Israel | 0.55 (4.43) | 0.02 (0.2) | 0.72 (5.76)*** |
| South Africa | 0.52 (4.16) | 0.48 (4.47)*** | 0.57 (5.75)*** |
| Saudi Arabia | 0.35 (2.82) | 0.51 (4.92)*** | 0.59 (5.79)*** |
| Kenya | 0.31 (2.5) | -0.07 (-0.59)*** | 0.89 (5.81)*** |
| Madagascar | 0.33 (2.64) | -0.04 (-0.36)*** | 0.89 (5.81)*** |
| Nigeria | 0.33 (2.37) | 0.44 (3.96)*** | 0.75 (5.81)*** |

$$f_T = 1.24(0.225***)f_{t-1} + \eta_t - 0.83(0.165***)\eta_{t-1}$$

Factor coefficients for countries synchronized with the global business cycle. Standard deviation in brackets.

*95% significance.

*** 99% significance.

one factor. We drop the countries with negative loadings and positive loadings below 0.1 and then we run the dynamic factor model to estimate the global business cycle indicator, only with data from the 49 synchronized countries. This grouping technique suggests synchronicity in the business cycle for developed economies, since, as we will point in Table 4, the 49 countries displayed in Table 3 are mainly highly globalized economies.

## 4.2. Convergence model coefficients

In this subsection we compare the results of the model for the two sets of estimations. Panels A and B from Table 4 show results from the convergence model estimations for the full sample

**Table 4. Convergence model coefficients, speed of convergence and tests.**

| | Panel A: Full sample (89) | | | | Panel B: Synchronized countries (49) | | | |
|---|---|---|---|---|---|---|---|---|
| | **(I)** | **(II)** | **(III)** | **(IV)** | **(V)** | **(VI)** | **(VII)** | **(VIII)** |
| $\ln(y_{it-1})$ | -0.019 (-7.94)** | -0.019 (-7.90)** | -0.038 (-13.15)** | -0.038 (-13.01)*** | -0.026 (-10.22)** | -0.026 (-10.25)** | -0.0384 (-12.53)** | -0.037 (-12.33)** |
| $\ln(KOF_{i,t})$ | --- | --- | 0.058 (11.76)** | 0.057 (11.54)*** | --- | --- | 0.038 (7.25)** | 0.035 (6.82)** |
| Global Business Cycle | --- | 0.003 (3.88)** | --- | 0.0026 (3.40)*** | --- | 0.005 (6.83)** | --- | 0.005 (6.57)** |
| Speed of Convergence λ (yearly %) | 0.04 | 0.04 | 0.08 | 0.08 | 0.06 | 0.06 | 0.09 | 0.08 |
| Wald Test (Joint significance)**** | $\chi^2(1)$ = 63.05 [.000] | $\chi^2(2)$ = 76.50 [.000] | $\chi^2(2)$ = 199.31 [.000] | $\chi^2(2)$ = 208.43 [.000] | $\chi^2(1)$ = 104.48 [.000] | $\chi^2(2)$ = 147.60 [.000] | $\chi^2(2)$ = 156.97 [.000] | $\chi^2(2)$ = 195.07 [.000] |
| F Test (indiv. Effects)**** | $F(88,3915)$ = 4.34 [.000] | $F(88,3914)$ = 4.37 [.000] | $F(88,3914)$ = 5.39 [.000] | $F(88,3913)$ = 5.39 [.000] | $F(48,2155)$ = 6.12 [.000] | $F(48,2154)$ = 6.36 [.000] | $F(48,2154)$ = 5.77 [.000] | $F(48,2154)$ = 5.98 [.000] |
| Hausman Test (Fixed vs. random Effects)**** | $\chi^2(1)$ = – [.000] | $\chi^2(2)$ = 53.38 [.000] | $\chi^2(2)$ = 83.05 [.000] | $\chi^2(3)$ = 117.82 [.000] | $\chi^2(1)$ = – [.000] | $\chi^2(2)$ = 82.92 [.000] | $\chi^2(2)$ = 103.15 [.000] | $\chi^2(3)$ = 101.45 [.000] |

*Note*: Panel data model: Fixed effects two stage least squares. AR (1) as instrumental variable (IV). Full sample and 49 countries synchronized with the global business cycle. In () t-statistics. In[] p-values.

* 90% significance

** 95% significance

*** 99% significance. Check Appendix 2 for test details.

and the synchronized countries, respectively. We run the model with the *ivpanel* function from *Matlab*, that estimates by OLS the beta coefficients in a two-stage setting. We select between fixed and random effects according to the results of the Hausman Test described below. Before the estimation, the instrumental variable $Z$ is constructed by introducing a one-period lag on the GDP levels explanatory variable.

Concerning synchronization, we find evidence of a greater effect on growth of the global business cycle variable for the set of countries whose cycles are more synchronized. This result can be extracted from the coefficients for the global business cycle variable in estimations IV and VIII from Table 4, our preferred specifications. The smaller coefficient of the global business cycle on the full sample estimation implies that when facing a global recession, asynchronized countries are less exposed to the propagation of the shock. However, synchronized countries show a stronger effect on GDP growth. With this result we confirm the hypothesis that synchronization improves convergence in global expansionary phases.

The global financial crisis of the late 2000s is a recent example, when less synchronized developing countries could weather relatively easier the negative global shock compared to the developed ones. This has translated into convergence between both groups of countries, as depicted in S1 Fig. For example, as [19] point out, regions such as Asia exhibited an unexpected resilience to the worst of the financial crisis. We contribute to this idea, arguing that desynchronization prevented these countries from contagion. But as synchronized countries tend to rank among the highly developed (globalized), it is straightforward to assume the asymmetry of this result: in the long run gains from synchronization are greater than risk in terms of economic growth. Consequently, and focusing on the variables upon study, developing countries can find opposing forces in the globalization process. More globalization tends to imply more synchronization (Table 2), and therefore it implies more risk of contagion from global crises. But in expansionary phases, more synchronization is positive for growth and therefore improves economic convergence, as can be interpreted from the greater column VIII coefficient (0.005) for the global business cycle variable compared to the column IV full sample coefficient (0.0026). [18] results support this: they find that when simulating shocks to a global factor the effects are stronger for countries more integrated in trade and/or financially.

In relation to globalization levels, the first conclusion that we can extract is that economic globalization is always positive for growth. This matches the theoretical assumption of the growth enhancing effects of economic integration. Of more interest to our study is the comparison of the globalization variable coefficients in columns IV and VIII, reflecting the greater effect of this variable for the full sample estimation (0.057) than for the subgroup of synchronized countries (0.035). This suggests that increasing globalization levels improve convergence. The main reason for the different coefficients from both samples is the way to run in terms of globalization that some of the countries from panel A present, which increases the average effect on GDP growth of a marginal increase in the average globalization level for this group. On the contrary, and as the data description from Table 2 shows, synchronized countries from panel B tend to be more globalized economies, and this reduces the marginal effect on growth of increasing their average level of globalization.

Therefore, our argument to explain the different effect on growth of the globalization levels from our two samples is that the synchronized countries group (panel B from Table 4) comprises mostly highly developed economies, and this means that globalization levels are close to maximum levels in these countries, so they present diminishing gains from increasing levels of globalization. However, panel A comprises a higher proportion of developing countries, which tend to be less globalized countries, and that increases the gains in terms of growth from increasing their levels of globalization.

Although the comparison of the speed of convergence (λ) of both panel A and B should be cautious since convergence inside both groups is related to different steady states, it is interesting to comment on the effect that the inclusion of the explanatory variables has on this parameter, and determine if it supports the argument of convergence between both groups of countries. Speed of convergence (and therefore $\beta_1$) for both groups increases substantially when including the globalization variable (compare I with III and V with VII) which indicates that, for a country to converge to the group steady-state (independently of if it is inside the synchronized or asynchronized), globalization is the key factor. Also, the almost equal speed of convergence in the baseline models (IV and VIII) implies a similar development towards steady-states in both groups, and a correct choice of the control variables. It is interesting to note that the speed of convergence is greater for the synchronized group when including as control only the global business cycle variable (II versus VI) which reinforces the idea that being synchronized is differential when explaining the different development patterns of both groups.

Comparing our results with previous studies, the positive effect of economic globalization on growth is in line with previous results (see [2, 8, 11]). As for synchronization, previous studies have documented the impact of global factors in determining variations of country GDP and convergence in business cycles, mainly for developed economies [10, 17], but to our knowledge, there is no paper estimating the effects of global business cycles on income convergence.

## 4.3. Robustness checks

In this section we test the robustness of our results, with two alternative specifications. In the first one, we repeat our baseline specification but changing the representation of the global business cycle: instead of the factor extracting the common behavior of country GDP cycle, we use World GDP growth rate series from the World Bank. Coefficients for these estimations are displayed in IX and X from Table 5. Although the coefficients for the newly introduced

**Table 5. Model coefficients for the alternative specifications.**

| First Alternative Specification (World GDP) | Panel A: Full sample (89) | Panel B: Synchronized countries (49) | Second Alternative Specification (Political KOF) | Panel A: Full sample (89) | Panel B: Synchronized countries (49) |
|---|---|---|---|---|---|
| | (IX) | (X) | | (XI) | (XII) |
| $\ln(y_{it-1})$ | -0.037 (-12.80)*** | -0.035 (-11.59)*** | $\ln(y_{it-1})$ | -0.033 (-11.51)*** | -0.037 (-11.75)*** |
| ln (KOF $_{i,t}$) | 0.059 (11.95)*** | 0.038 (7.48)*** | ln (political KOF $_{i,t}$) | 0.044 (8.97)*** | 0.034 (5.92)*** |
| Global Business Cycle (World GDP growth rate*) | 0.002 (3.62)*** | 0.003 (6.47)*** | Global Business Cycle (factor) | 0.003 (3.62)*** | 0.005 (6.75)*** |
| Third Alternative Specification (GDFM factor) | Panel A: Full sample (89) | Panel B: Synchronized countries (49) | | | |
| | (XIII) | (XIV) | | | |
| $\ln(y_{it-1})$ | -0.037 (-12.93)*** | -0.024 (-4.79)*** | | | |
| ln (KOF $_{i,t}$) | 0.059 (11.82)*** | 0.025 (3.84)*** | | | |
| Generalized dynamic factor model | 0.002 (2.70)*** | 0.001 (1.36) | | | |

*Note*: Panel data model: Fixed effects two stage least squares. AR (1) as instrumental variable (IV). Full sample and 49 countries synchronized with the global business cycle. In () t-statistics.

* 90% significance

** 95% significance

*** 99% significance.

variable are smaller than the ones in IV and VIII (our preferred specifications) the intuition behind its interpretation remains unaltered: synchronized countries are more affected by fluctuations in the global economy, but using the World GDP growth rate series as explanatory this effect is less relevant. We include this variable in the robustness check instead of in our baseline model since we consider that the definition of synchronization and global fluctuations are better captured if constructed from the GDP data that generates the coefficients of the model, instead of other data sources which could generate unwanted effects in the results. S4 Fig shows the similar behavior of World GDP Series compared to the global business factor used in the baseline estimation.

The second alternative specification is referred as another proxy for globalization. Instead of using the economic component of the KOF index, we use the political component. It is composed by the following indicators, for every country: by the number of embassies, the number of memberships in international organizations and the number of UN peace missions. Coefficients are displayed in columns XI and XII from Table 5. In this case, we again find a similar pattern to the one observed in our baseline scenario, since synchronized countries are less affected by changes in globalization. Again, the difference between panel A and B coefficients is smaller than in the preferred specification, which suggests that changes in economic globalization are more relevant than changes in political globalization when explaining convergence between both groups.

In the third specification, we construct the factor with a different methodology (GDFM, see [37] for details). This explanatory variable is not significant in panel B estimation (column XIV). This is probably due to the fact that we have estimated the factor with the subsample of the 49 countries selected in the baseline specification (in order to keep the same samples for easiness of comparison), which could alter the composition of the estimated factor and also its correlation with the previously selected synchronized countries. The rest of coefficients remain in line with the estimated in previous settings.

## 5. Conclusions

Our methodology introduces two main novel aspects to the economic convergence literature. First, the variables explaining convergence. A factor replicating a global business cycle is used as an explanatory variable to capture the effect of global shocks on country specific GDP growth. Synchronized countries are those who benefit (suffer) more from the effects of global expansions (recessions). On the contrary, asynchronized countries will benefit less from global expansions, and will be less affected by global recessions. The significant coefficients of the global business cycle on both sets of parameters imply that global shocks have long term effects on country GDP growth for developing and developed countries, but the effect is greater for the latter group. Second, the technique used to group countries with the criteria of interest proposed. In this respect, the methodological novelty (countries synchronized with the factor), has been also useful for describing a new pattern of conditional convergence in the literature.

We also show that, within each country, globalization levels and GDP per capita growth are positively correlated. For synchronized countries, globalization has a relatively smaller impact on growth. These countries tend to be more globalized, therefore, the impact of a marginal increase of globalization levels is less relevant for growth. But the accumulated effect of the globalization process they have experienced, during decades or centuries, explains the difference in GDP and globalization levels between both groups of countries.

The previous reasoning implies that a higher level of globalization is always positive for growth, but its marginal effect is decreasing. At the same time, as the marginal effect of the globalization level on growth is reduced, these countries can expect to be more synchronized,

which may in turn be positive (during global expansions) for growth and therefore for economic convergence. As our subgroup of synchronized countries is composed of highly globalized economies, our results also point to the idea that closer economic integration tends to synchronize cycles across countries. To sum it up, in the economic development process, countries should expect decreasing effects of globalization levels on growth, but increasing effects on growth coming from the synchronization with the global business cycle.

Some limitations of the paper appear in the intertemporal comparison of the results, the extension of the data, the measurement of convergence, the methodology and the definition of globalization. For the first point, the dynamics of income convergence will eventually synchronize more countries with the global business cycle, which makes our results just useful for a given period of time. As the definition of the factor will in future estimations also change since it is composed by country GDP data, this will also alter the results and make them more difficult to compare with estimations of previous years. In this respect, the synchronized and asynchronized groups will also vary in composition. Second, the extension of the data to include more countries and higher frequency data could also be addressed in future versions of this paper. Third, more than one indicator of convergence could be included to strengthen the validity of the results. Fourth, using other estimator in the dynamic panel, for example the generalized method of moments (GMM) [38] or other methodology such as a Bayesian VAR (BVAR). Fifth, the definition of globalization could be extended to include other aspects such as migration, knowledge diffusion or cultural distance/proximity.

In terms of economic policy, we prove that more globalization fosters growth, and this should be taken into account for catching up countries. Synchronization is an almost definitive economic consequence for countries that reach a sufficient level of globalization, as explained in detail in section 3 of this paper. But before this happens, it is needed to develop policies aiming to enhance economic integration with the rest of the world. Developing countries benefit more from increasing levels of globalization, therefore they should have a stronger incentive to implement these policies. But these should be implemented with caution, as more integration also tends to imply a higher degree of synchronization with the global cycle, which may entail risks from contagion [39] during global recessions.

Further research on this topic and its relation with growth of GDP may explore the different channels of economic globalization (trade, finance or migration) as explanatory variables and simulations of shocks to these explanatory variables. Additionally, the effects of global, regional and idiosyncratic cycle components on country GDP fluctuations or the grouping of countries by income level subsamples, sharing different regional business cycles or with a different grouping methodology to the one used in this paper. Lastly, another aspect that would be interesting to analyze is the implications of the last decade economic globalization process slowdown, which together with the COVID-19 recession may alter significantly some of the conclusions of this work if the datasets are extended to include current years. In this respect, a possible change in economic integration is to shift to regions in the coming years instead of being so dependent on global integration, and this could have an effect on income convergence patterns [40].

## Appendix 1

### Full panel of countries (89)

Argentina, Australia, Austria, Barbados, Belgium, Brazil, Bulgaria, Canada, Colombia, Costa Rica, Cyprus, Denmark, Finland, France, Germany, Greece, Guatemala, Hong Kong, Hungary, Iceland, Indonesia, Ireland, Israel, Italy, Jamaica, Japan, Kenya, Luxembourg, Madagascar, Malaysia, Mexico, Netherlands, New Zealand, Nigeria, Norway, Philippines, Portugal,

Romania, Saudi Arabia, Singapore, South Africa, Spain, Sweden, Switzerland, Thailand, Trinidad and Tobago, United Kingdom, United States, Venezuela, Malta, Albania, Polonia, Turkey, Cambodia, China, India, Myanmar, Pakistan, Korea, Dem. Rep., Vietnam, Bolivia, Chile, Dominican Republic, Ecuador, Peru, Uruguay, Iraq, Jordan, Kuwait, Syrian Arab Republic, Yemen, Rep., Algeria, Burkina Faso, Cameroon, Cote d'Ivoire, Congo, Dem. Rep., Egypt, Arab Rep., Ethiopia, Ghana, Malawi, Mali, Mauritania, Niger, Senegal, Sudan, Tanzania, Tunisia, Uganda, and Zambia.

### Synchronized countries (49)

Argentina, Australia, Austria, Barbados, Belgium, Brazil, Bulgaria, Canada, Colombia, Costa Rica, Cyprus, Denmark, Finland, France, Germany, Greece, Guatemala, Hong Kong, Hungary, Iceland, Indonesia, Ireland, Israel, Italy, Jamaica, Japan, Kenya, Luxembourg, Madagascar, Malaysia, Mexico, Netherlands, New Zealand, Nigeria, Norway, Philippines, Portugal, Romania, Saudi Arabia, Singapore, South Africa, Spain, Sweden, Switzerland, Thailand, Trinidad and Tobago, United Kingdom, United States, Venezuela and Malta.

### Asynchronized countries (40)

Albania, Polonia, Turkey, Cambodia, China, India, Myanmar, Pakistan, Korea, Dem. Rep., Vietnam, Bolivia, Chile, Dominican Republic, Ecuador, Peru, Uruguay, Iraq, Jordan, Kuwait, Syrian Arab Republic, Yemen, Rep., Algeria, Burkina Faso, Cameroon, Cote d'Ivoire, Congo, Dem. Rep., Egypt, Arab Rep., Ethiopia, Ghana, Malawi, Mali, Mauritania, Niger, Senegal, Sudan, Tanzania, Tunisia, Uganda, and Zambia.

## Appendix 2

### Tests

We run a set of relevant tests to validate the estimates of our dynamic panel data model. Table 6 presents tests coefficients for every estimation, but in this section we focus only on the test results for the relevant (IV and VIII) estimations of this paper. Tests are run in *Matlab* with the panel data toolbox developed by [34]. Details on the foundations and rationale of the Wald and Hausman tests can be found in [41].

**Wald test.** We test the joint significance of the coefficients of every estimation using the chi-square distribution Wald test. Under the null hypothesis we test if a given value multiplying the independent variables is zero: that is why it is called the linear hypothesis test. According to the low p-values for the statistics of the test performed for IV (208.43) and VIII (195.07) estimations, with two degrees of freedom we reject the null at the 95% significance for the two relevant estimations performed, which implies joint significance for both sets of coefficients.

**Hausman test.** We run the Hausman Test to test for consistency of the fixed effects estimations. Under the alternative hypothesis, only the random effects model would result in

**Table 6. Main explanatory variables.**

| Name | Source | Description |
|---|---|---|
| KOF Index | KOF Swiss Economic Institute | • Economic, political and social globalization index |
| GDP | Conference Board | • GDP growth |
| Global business cycle | Own estimation | • Dynamic factor (replicating global business cycle) |
| World GDP* | World Bank | • World GDP growth |

*Used in the robustness check estimations.

consistent and efficient estimators. This test is convenient to choose between fixed or random effects for our estimations, since our $T$ is small relative to $i$. On large $T$ samples fixed and random effects estimators tend to be similar. According to the values of the chi-square distribution Hausman Tests for IV (117.82) and VIII (101.45) with the corresponding (3) degrees of freedom, the fixed effects estimations are not biased and consistent. The very small *p-values* for the tests performed imply that we reject the null hypotheses of the existence of non-systematic differences on the fixed and random effects estimators, which imply that we should choose the more consistent estimators, in this case, the fixed effects.

**F test for individual effects.**   Since we have chosen fixed effects estimators for our model according to the Hausman tests results, we need to test the existence of individual time invariant individual effects affecting the model. The very significant coefficients for the *Chow F test* for the set of parameters IV (5.39) and VIII (5.98) imply that we reject the null of the non-existence of individual effects, which reinforces the use of a fixed effects (within estimators) model.

## Supporting information

**S1 Fig. GDP convergence for synchronized and asynchronized countries.** Source: Indexes are own elaboration (Base year is 1950 = 100) from GDP data from the Conference Board Total Economy Database. Synchronized countries are those correlated with the factor explaining global economy fluctuations.
(TIF)

**S2 Fig. Economic globalization for high, middle (upper and lower) and low income countries.** Data source: KOF Swiss Economic Institute. Graph is own elaboration.
(TIF)

**S3 Fig. Economic globalization and synchronization (synchronized countries).** Source: Own elaboration from KOF index and own estimates for synchronization.
(TIF)

**S4 Fig. Dynamic factor vs world GDP.** Source: Own elaboration for the dynamic factor and World Bank for World GDP Series.
(TIF)

## Author Contributions

**Conceptualization:** Maria Jesus Delgado.

**Data curation:** Andres Lopez, Sonia De Lucas.

**Formal analysis:** Andres Lopez, Maria Jesus Delgado.

**Investigation:** Andres Lopez.

**Methodology:** Sonia De Lucas.

**Project administration:** Maria Jesus Delgado.

**Software:** Andres Lopez, Sonia De Lucas.

**Supervision:** Sonia De Lucas, Maria Jesus Delgado.

**Writing – original draft:** Andres Lopez.

**Writing – review & editing:** Andres Lopez.

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
