## [Decision Letter · Decision Letter 0]

9 Jun 2021

PONE-D-21-12597

Economic Convergence in a Globalized World: The Role of Business Cycle Synchronization

PLOS ONE

Dear Dr. Lopez,

Thank you for submitting your manuscript to PLOS ONE. After careful consideration, we feel that it has merit but does not fully meet PLOS ONE’s publication criteria as it currently stands. Therefore, we invite you to submit a revised version of the manuscript that addresses the points raised during the review process.

Note that both referee's have included further comments to the ones below in the attached documents.

We look forward to receiving your revised manuscript.

Kind regards,

Juan Carlos Cuestas

Academic Editor

PLOS ONE

2. We note that S2 Figure and S3 Figure in your submission contain map images which may be copyrighted. All PLOS content is published under the Creative Commons Attribution License (CC BY 4.0), which means that the manuscript, images, and Supporting Information files will be freely available online, and any third party is permitted to access, download, copy, distribute, and use these materials in any way, even commercially, with proper attribution. For these reasons, we cannot publish previously copyrighted maps or satellite images created using proprietary data, such as Google software (Google Maps, Street View, and Earth). For more information, see our copyright guidelines: http://journals.plos.org/plosone/s/licenses-and-copyright.

a. You may seek permission from the original copyright holder of Figure(s) [#] to publish the content specifically under the CC BY 4.0 license. 

Additional Editor Comments (if provided):

Reviewers' comments:

Reviewer's Responses to Questions

**Comments to the Author**

1. Is the manuscript technically sound, and do the data support the conclusions?

Reviewer #1: Partly

Reviewer #2: Yes

2. Has the statistical analysis been performed appropriately and rigorously? 

Reviewer #1: No

Reviewer #2: Yes

3. Have the authors made all data underlying the findings in their manuscript fully available?

Reviewer #1: Yes

Reviewer #2: Yes

4. Is the manuscript presented in an intelligible fashion and written in standard English?

Reviewer #1: Yes

Reviewer #2: Yes

5. Review Comments to the Author

Reviewer #1: The manuscript aims to explore the effects of economic globalization on economic convergence for a sample of 89 developed and developing countries in the period 1970-2015 by means of a dynamic panel data model. Overall, the paper approaches a hotly debated topic, being well structured and properly documented. At very first glance, the major concern is depicted by the quantitative approach which is too common. Nevertheless, there occur several shortcomings that should be addressed, as detailed in the attache Review Report.

Reviewer #2: ECONOMIC CONVERGENCE IN A GLOBALIZED WORLD: THE ROLE OF BUSINESS CYCLE SYNCHRONIZATION.

To begin with, the authors provide a historical context on the literature of economic growth and convergence. They suggest that the strand of literature studying economic convergence has concentrated on analyzing the converging process itself, rather than locating the causes that enforce this process, despite the continuous effort to identify the drivers of economic growth. They argue that the global financial crisis of 2008 and the COVID-19 recession are events that highlight the importance of economic integration-specifically in the context of economic convergence. Furthermore, they indicate that in an increasingly globalized world, business cycle synchronization has a twofold role: first, the synchronization of business cycles should be a policy aim in the effort to increase economic integration, and second, synchronization along with integration are significant aspects for economic convergence. In that line, they suggest that these two are the main drivers of economic convergence. Their work aims to prove that business cycle synchronization improves economic convergence in expansionary phases but exposes the countries to the risk of contagion in recessionary phases. Moreover, they propose that globalization-inducing policies in a-synchronized countries have greater effects on growth.

Following Artis and Ocubo (2008), the authors adopt the notion that globalization is the participation of many countries in the formation of a global business cycle. They associate the increased economic integration with more synchronicity of the world business cycle. Their literature review, therefore, is focused on discussing prior work of two main areas of interest: namely globalization and growth and the global business cycle and growth. Previous works studying the effect of globalization on growth, have been using as proxies for globalization either trade or financial and labor markets. They argue that even though those three are the main channels that promote globalization, the concept is wider and involves political and institutional factors. Therefore, either of those three proxies fails to capture the broader effect of globalization on growth. To that extent, indexes proxying for globalization have reported better results.

While discussing previous work on business cycles and growth, two seemingly contradictory views concerning the cause of economic fluctuations that impact growth emerge. They propose that the intuition has traditionally been that domestic shocks are those that impact growth. However, more recent studies suggest that the causes of these domestic shocks can be attributed to fluctuations of the world business cycle, and that is the hypothesis they adopt.

While discussing previous work on business cycles and growth, two seemingly contradictory views emerge, concerning the cause of economic fluctuations that impact growth. They propose that the intuition has traditionally been that domestic shocks are those that impact on growth. However, more recent studies suggest that the cause of these domestic shocks can be attributed in fluctuations of the world business cycle, and that is the hypothesis they prescribe to. Despite the theoretical allure of this hypothesis, the authors point out potential problems in the empirical use of global business cycles in models. The main problems are the definition of global recessions and data accuracy. The definition of global recessions is unclear, a fact that is highlighted in the construction of the global business cycle. Since negative growth rates are negated by positive growth rates in other countries, global economic recessions are hard to pinpoint. Moreover, data are not accurate for all countries across time.

A preliminary analysis of the data provides two important stylized facts. The distribution of countries according to the globalization levels shows that the more globalized countries are also the more synchronized. On the contrary, 80% of the asynchronized countries are among the less globalized. To that extent, the asymmetric effect between globalization and synchronization is pointed out. The relationship between globalization and synchronization is more prevalent among the less globalized and asynchronized countries. This leads to the second fact; the authors conclude that globalization is a necessary condition for economic convergence. Lastly, the authors point to the dynamic relationship between synchronization and economic growth, with synchronized countries experiencing more growth during expansions and more risk of contagion during economic recessions.

The problems described above, that arise from the nature of globalization and the effort to synthesize a global business cycle, inform to an important extent the methodological choices of the authors. The use of the KOF globalization index is more economical than the use of several explanatory variables and has reported better results in the literature. Moreover, the authors employ a dynamic factor approach as a grouping technique, to filter the synchronized countries and then simulate the world business cycle according to the common trend presented in their data.

The dynamic factor model is employed, and Bai and Ng’s criteria is used to certify the existence of one factor. The countries reporting positive factor loading are then selected to simulate the business cycle, which is comprised finally by 49 countries.

After the construction of the global business cycle, the authors proceed to estimate a panel data convergence model, where GDP growth is regressed in the common factor that proxies for the global business cycle and the globalization index. The regression equation is the following.

ln(y_(i,t)/y_(i,t-1) )=β_0+β_1 ln(y_(i,t-1) )+β_2 (〖factor〗_(i,t) )+β_3 (〖KOF〗_(i,t) )+α_i+u_(i,t)

α_i is the country-specific fixed effects mean, while β_2 and β_3 represent the effects of synchronization and globalization, respectively. Due to the potential endogeneity and reverse causality the model is estimated using two-stage fixed effects, with the lag of the dependent variable acting as an instrument. This model is estimated using for the whole sample of 89 countries, and a sub-sample that contains the highly synchronized countries, in 4 different specifications.

The results suggest that both globalization and business cycle synchronicity have a positive impact on growth. Furthermore, globalization has a greater effect on the full sample of countries, which suggests that its effects on growth is decreasing on more synchronization. This means that the hypotheses are both confirmed. Finally, the specification including both explanatory variables report higher speeds of convergence. The process is repeated using another aspect of the composite globalization index, KOF, to proxy for globalization and the World GDP growth rate to proxy for the global business cycle. The qualitative qualities of the results are replicated even though the effects are smaller.

 

Comments - Questions

The causal relationship between synchronization, globalization, and growth. The authors hypothesize that business cycle synchronization is a facet of globalization. I have not understood whether this view is expressed as considering the synchronization as a prerequisite via policy channels, or whether it is viewed as a natural and unavoidable process.

An interesting point is that globalization implies higher growth levels but a greater risk of contagion, as well. However, the symmetry of this result is not discussed, i.e., it is not shown whether the gains from globalization in terms of growth are generally greater than the losses during the recessionary phases.

The discussion of the stylized facts in section 3.1 could be further supported via a scatter plot of the countries according to globalization and synchronization. The clusters that would form on the scatter plot would also be among the countries comprising the global business cycle.

In section 4.1 in the factor loading equation, is there a typo in η_(t-1)?

It would be interesting to include a graph presenting the evolution of the global business cycle in time and compare it with the World GDP growth used in the robustness tests to replicate the results. Moreover, it was unclear whether the global business cycle managed to reflect recessionary phases as well.

In section 5, in the last paragraph it is suggested that the last decade of globalization process, along with the recent COVID recession may alter some of the conclusions of the present work. However, considering the recent recession and its specific character, due to the disruption of global supply chains, wouldn’t we expect a validation of the results? After all, it is argued that globalization implies risks during recessions.

Would it be possible to include, the list of the countries in the sample in an appendix? While the list of synchronized countries is included in the report of the factor loadings, the extended sample is not outlined.

The text had a few grammatical errors. I have commented on them on a separate word file, on the text. I am hoping it is not out of place.

See whether you can update the literature. See for instance:

Holmes, M., Otero, J. and Panagiotidis, T., (2014), A NOTE ON THE EXTENT OF U.S. REGIONAL INCOME CONVERGENCE, Macroeconomic Dynamics, 18, issue 7, p. 1635-1655.

6. PLOS authors have the option to publish the peer review history of their article (what does this mean?). If published, this will include your full peer review and any attached files.

Reviewer #1: No

Reviewer #2: No

---

## [Author Response · Author response to Decision Letter 0]

22 Jul 2021

To the editor and reviewers:

To this letter we attach an updated version of the paper. Here we respond to the suggestions from the academic editor and both reviewers. In our answers to reviewers we specify where exactly in the text we have added the changes. We appreciate the comments, since we have improved substantially the content and structure of the paper. 

RESPONSE TO ACADEMIC EDITOR

1. Please ensure that your manuscript meets PLOS ONE’s style requirements, including those for file naming. The PLOS ONE style templates can be found at

We changed the format of titles and subtitles, and file naming style (title of the paper).

2. We note that S2 Figure and S3 Figure in your submission contain map images which may be copyrighted. We require you to either (1) present written permission from the copyright holder to publish these figures specifically under the CC BY 4.0 license, or (2) remove the figures from your submission. 

We removed maps and plotted economic globalization component from KOF index. Own elaboration graphs, by income level groups (Figure 2). Additionally, we included new Figure 3 (scatter plot) and Figure 4 (factor and World GDP series plotted together) as suggested by the reviewers.

RESPONSE TO REVIEWERS

Reviewer 1:

1.The introductory section presents in a suitable manner the background of the paper, but there is missing the gap in the literature. Which is the novelty and originality of current study? How does present manuscript contribute to the existing literature? Besides, the purpose of the research should be deeply argued. 

We have extended in the introduction section on novelty, contribution to existing literature and research purpose. 

2. The section dedicated to earlier literature should be expanded with a table summarizing prior studies. As well, the author(s) should extend the discussion of existing studies on beta-convergence methodology, along with stochastic framework. 

Table at the end of literature review. Beta convergence and alternative convergence methodologies section at the end of literature review.

3. With reference to data and methodology, I recommend the author(s) to develop a table regarding the selected variables. 

Done. End of section 3.

4. As well, I suggest the use of Bayesian panel VAR techniques as baseline approach. Besides, the robustness should be checked through Markov-switching dynamic factor model (MS-DFM) proposed by Diebold and Rudebusch (1996) and Kim and Yoo(1995). 

We consider the Dynamic panel data model the best methodological approach to study income convergence, since it gives us flexibility to add explanatory variables in the context of comparison among group different groups of countries. Additionally, data panel is the methodology used in papers with similar structure to ours (Dreher; Villaverde; Samimi) and we found useful to use a similar methodology, in order to facilitate comparison of results with previous research. The BVAR approach, to our knowledge has not been used in an income convergence setting or to calculate beta convergence, but we find interesting the suggestion and will try to include it in future research as alternative specification to the baseline. We have already found Matlab scripts https://sites.google.com/site/dimitriskorobilis/matlab/code-for-vars to develop the BVAR model that could fit in an income convergence model. We have in fact mentioned the BVAR in the limitations in the conclusions section, as a possible extension of the research. 

For the suggestion of the factor, we use the GDFM (Generalized Dynamic Factor Model) procedure in a third specification in the robustness checks. M. Forni, M. Hallin, M. Lippi, P. Zaffaroni (2017) “Dynamic Factor Models with infinite-dimensional factor space: Asymptotic analysis”, Journal of Econometrics, 199, 74-92

5. In addition, public governance measures should be comprised in the econometric framework. 

We include the political component of the KOF index in the robustness checks. We have extended the definition of this component of the KOF index in second paragraph of section 4.3. 

6. With respect to the empirical outcomes’, the author(s) should compare the results with prior studies. Therewith, the net directional network connectedness should be reported. 

Last paragraph in section 4.2. compares the results with prior studies.

7. The sub-section 4.3 is too straightforward and should not be disclosed 

ok, in appendix. 

8. The last section should focus more on policy/practical implications. As well, study limitations should be noticed. 

We comment on more policy implications and also limitations in section 5.

Reviewer 2:

1.The causal relationship between synchronization, globalization, and growth. The authors hypothesize that business cycle synchronization is a facet of globalization. I have not understood whether this view is expressed as considering the synchronization as a prerequisite via policy channels, or whether it is viewed as a natural and unavoidable process. 

We explain in the fifth paragraph of section 5 (conclusions) our view on synchronization as a natural process, for countries that apply economic policy to enhance globalization.

2. An interesting point is that globalization implies higher growth levels but a greater risk of contagion, as well. However, the symmetry of this result is not discussed, i.e., it is not shown whether the gains from globalization in terms of growth are generally greater than the losses during the recessionary phases.

 We explain the asymmetry of our results in section 4.2, in the middle of paragraph 3. 

3. The discussion of the stylized facts in section 3.1 could be further supported via a scatter plot of the countries according to globalization and synchronization. The clusters that would form on the scatter plot would also be among the countries comprising the global business cycle. 

See Figure 3. Thanks for the suggestion.

4. In section 4.1 in the factor loading equation, is there a typo in η_(t-1)? 

We have corrected the typo.

6. It would be interesting to include a graph presenting the evolution of the global business cycle in time and compare it with the World GDP growth used in the robustness tests to replicate the results. 

See figure 4 for plot of World GDP and the global business cycle estimated.

7. Moreover, it was unclear whether the global business cycle managed to reflect recessionary phases as well. 

Yes, it does. See Figure 4 again where co-movement of both series is depicted. 

8. In section 5, in the last paragraph it is suggested that the last decade of globalization process, along with the recent COVID recession may alter some of the conclusions of the present work. However, considering the recent recession and its specific character, due to the disruption of global supply chains, wouldn’t we expect a validation of the results? After all, it is argued that globalization implies risks during recessions. 

We comment it on the last line of the paper. 

9. Would it be possible to include, the list of the countries in the sample in an appendix? While the list of synchronized countries is included in the report of the factor loadings, the extended sample is not outlined. 

Thank you for the suggestion, we have included Appendix 1. Lists of synchronized, asynchronized y full sample of countries.

10. The text had a few grammatical errors. I have commented on them on a separate word file, on the text. I am hoping it is not out of place. 

We accepted all the grammatical changes proposed and reviewed the text for further improvements in the writing. Thank you.

11. See whether you can update the literature. See for instance:

Holmes, M., Otero, J. and Panagiotidis, T., (2014), A NOTE ON THE EXTENT OF U.S. REGIONAL INCOME CONVERGENCE, Macroeconomic Dynamics, 18, issue 7, p. 1635-1655. 

Included in the last paragraph of 2.3, in the extension of the reviewed literature on income convergence.

---

## [Decision Letter · Decision Letter 1]

2 Aug 2021

Economic convergence in a globalized world: The role of business cycle synchronization

PONE-D-21-12597R1

Dear Dr. Lopez,

We’re pleased to inform you that your manuscript has been judged scientifically suitable for publication and will be formally accepted for publication once it meets all outstanding technical requirements.

Kind regards,

Juan Carlos Cuestas

Academic Editor

PLOS ONE

Additional Editor Comments (optional):

Reviewers' comments:

Reviewer's Responses to Questions

**Comments to the Author**

1. If the authors have adequately addressed your comments raised in a previous round of review and you feel that this manuscript is now acceptable for publication, you may indicate that here to bypass the “Comments to the Author” section, enter your conflict of interest statement in the “Confidential to Editor” section, and submit your "Accept" recommendation.

Reviewer #1: All comments have been addressed

Reviewer #2: All comments have been addressed

2. Is the manuscript technically sound, and do the data support the conclusions?

Reviewer #1: Yes

Reviewer #2: Yes

3. Has the statistical analysis been performed appropriately and rigorously? 

Reviewer #1: Yes

Reviewer #2: Yes

4. Have the authors made all data underlying the findings in their manuscript fully available?

Reviewer #1: Yes

Reviewer #2: Yes

5. Is the manuscript presented in an intelligible fashion and written in standard English?

Reviewer #1: Yes

Reviewer #2: Yes

6. Review Comments to the Author

Reviewer #1: The revised version of the manuscript improved in a proper manner. The whole suggestions and recommendations were implemented suitably. As well, appropriate explanations were provided.

Reviewer #2: (No Response)

7. PLOS authors have the option to publish the peer review history of their article (what does this mean?). If published, this will include your full peer review and any attached files.

Reviewer #1: No

Reviewer #2: No

---

## [Editor Report · Acceptance letter]

6 Aug 2021

PONE-D-21-12597R1 

Economic convergence in a globalized world: The role of business cycle synchronization 

Dear Dr. Lopez:

I'm pleased to inform you that your manuscript has been deemed suitable for publication in PLOS ONE. Congratulations! Your manuscript is now with our production department. 

Kind regards, 

on behalf of

Professor Juan Carlos Cuestas 

Academic Editor

PLOS ONE